# Influence of Salinity on the Microbial Community Composition and Metabolite Profile in Kimchi

**Mi-Ai Lee [1]** , Yun-Jeong Choi [1], Hyojung Lee [1], Sojeong Hwang [1], Hye Jin Lee [1], Sung Jin Park [1] , Young Bae Chung [1], Ye-Rang Yun [1] , Sung-Hee Park [1], Sunggi Min [1], Lee-Seung Kwon [2],* and Hye-Young Seo [1],*

[1]  Kimchi Industry Promotion Division, World Institute of Kimchi, Gwangju 61755, Korea; leemae@wikim.re.kr (M.-A.L.); yjchoi85@wikim.re.kr (Y.-J.C.); zxgod0718@hanmail.net (H.L.); ghkdthwjd7@gmail.com (S.H.); hjl0317@gmail.com (H.J.L.); parksungjin0000@gmail.com (S.J.P.); ybchung@wikim.re.kr (Y.B.C.); yunyerang@wikim.re.kr (Y.-R.Y.); shpark@wikim.re.kr (S.-H.P.); skmin@wikim.re.kr (S.M.)

[2]  Department of Health Care Management, Catholic Kwangdong University, Gangneung-si 25601, Korea

*  Correspondence: leokwon1@cku.ac.kr (L.-S.K.); hyseo@wikim.re.kr (H.-Y.S.); Tel.: +82-33-649-7274 (L.-S.K.); +82-62-610-1731 (H.-Y.S.)

**Abstract:** Kimchi, a popular traditional Korean fermented food, is produced by fermenting vegetables with various spices and salt. Salt plays an important role in the preparation of kimchi and affects its taste and flavor. This study aimed to investigate the effects of salinity on kimchi fermentation. The salinities of five sets of kimchi samples were adjusted to 1.4%, 1.7%, 2.0%, 2.2%, and 2.5%. The characteristics of each kimchi sample, including its pH, acidity, free sugar content, free amino acid content, organic acid content, and microbial community composition, were evaluated during kimchi fermentation. The low-salinity kimchi sample showed a rapid decline in the pH at the beginning of the fermentation process, a relatively high abundance of *Leuconostoc mesenteroides*, and high mannitol production. In the late fermentation period, *Latilactobacillus sakei* had a higher abundance in the kimchi sample with high salinity than in other samples. In the initial stage of fermentation, the metabolite composition did not differ based on salinity, whereas the composition was considerably altered from the third week of fermentation. The findings showed variations in the characteristics and standardized manufacturing processes of kimchi at various salt concentrations. Therefore, salinity significantly affected the types and concentrations of fermentation metabolites in kimchi.

**Keywords:** kimchi; salinity; fermentation; lactic acid bacteria; metabolites

## 1. Introduction

Kimchi is one of the most popular traditional fermented foods in Korea that is prepared using lactic acid bacteria (LAB) fermentation [1]. The type and concentration of various components and the fermentation temperature affect the flavor, taste, and characteristics of kimchi [2–4]. Several studies have suggested that fermentation can affect the bioactive compounds of kimchi and LAB profiles, which can significantly alter the sensory and nutritional qualities of kimchi [5–7]. However, even though salt is one of the most important factors in fermentation, there is limited knowledge on the effects of salt on the quality of kimchi [8,9].

Salt is an important influencing factor in most fermented foods, including kimchi, as it extends the shelf life of the food and improves its taste [10]. Salt is generally associated with the physical and sensory properties and preservation of fermented foods [11]. However, several studies have suggested that high salt intake from dietary sources is strongly associated with hypertension and cardiovascular disease [12–14]. Salt helps maintain the quality of kimchi by eliminating harmful bacteria and promoting LAB colonization, which is the most important process in kimchi fermentation [15]. The type and concentration of salt affect the changes in the microbial community composition and metabolite profile during kimchi fermentation [16,17]. The previous study has reported that changes

in bacterial profiles due to different salt treatments led to changes in kimchi metabolite profiles in relation to sensory and nutritional quality [8]. Although several studies have characterized the microbial community in kimchi through a metagenomics approach, there are few studies on the effects of salt concentration on the differences in the metabolites produced by microorganisms in kimchi [9,16]. The salt concentration (0, 1.25, 2.5, 5%) suggested in the previous study [9] is for research purposes, and no kimchi is actually prepared in this manner. There is no commercial kimchi that has such a high concentration of salt (5% salinity of kimchi). Our preliminary study for monitoring the salinity of kimchi distributed on the market revealed that the salinity of kimchi was 1.4~2.5%, and the salinity of kimchi distributed in Korea, China, and United States were in the range of 2.0~2.3% [18]. In addition, Hong et al. [16] analyzed the change in the microbial profiles according to the salt concentration of kimchi but did not analyze the correlation between metabolites and microbial community. Since many microorganisms present in the fermentation stage of kimchi contribute to metabolite production, it is difficult to establish a clear relationship between salt concentration and the metabolite profile in kimchi.

This study primarily aimed to investigate the effects of salinity on kimchi fermentation. The comparison of the microbial community composition and metabolite profiles at five different salinities (1.4%, 1.7%, 2.0%, 2.2%, and 2.5%) is an important strategy for a holistic understanding of the process of kimchi fermentation.

## 2. Materials and Methods

### 2.1. Experimental Materials

Kimchi cabbage and garlic were purchased from the western agricultural and fishery market in Gwangju (Republic of Korea). Red pepper powder (Geumchi, Gwangju, Korea), pure salt (Hanju salt, Ulsan, Korea), and other ingredients (including white sugar from Samyang Corporation, Ulsan, Korea) were purchased from the same market. All experimental analyses were performed using analytical-grade reagents obtained from Daejung (Gyeonggi-do, Korea). Water and acetonitrile used for high-performance liquid chromatography (HPLC) were of chromatographic grade (Merck, Kenilworth, NJ, USA).

### 2.2. Preparation of Kimchi

The kimchi cabbage was soaked in 10% ($w/v$) pure salt solution for 18 h. Subsequently, the salted kimchi cabbage was washed three times with water and then drained for 2 h. The salted kimchi cabbage was cut into 3 cm × 3 cm pieces to prepare kimchi. The seasoning mixture was prepared by mixing 29.4% radish, 10.6% scallion, 13.20% red pepper powder, 7.2% crushed garlic, 2.03% crushed ginger, 5.73% fermented fish sauce, 10% vegetable broth, 3% sugar, and 11.47% glutinous rice paste. The mixture was then added to the salted kimchi cabbage at a ratio of 70:30 (salted cabbage:seasoning mixture). The final salinity of the kimchi was adjusted to 1.4%, 1.7%, 2.0%, 2.2%, and 2.5% (1.4%, SK-A; 1.7%, SK-B; 2.0%, SK-C; 2.2%, SK-D; 2.5%, SK-E). Each kimchi sample (600 g) was separately packed in a polyethylene film and sealed using a vacuum packaging machine (AZC-070, INTRISE, Ansan, Korea). The packed kimchi was stored for 5 weeks in a refrigerator at 6 °C, and its characteristics were analyzed at intervals of 1 week.

### 2.3. Microbial Community Analysis

Total DNA was extracted from the samples using a PowerSoil DNA Isolation Kit (Mo Bio Laboratories, Carlsbad, CA, USA) according to the manufacturer's instructions. DNA concentration and purity were measured using a NanoDrop ND-2000 (Thermo Fisher Scientific Inc., Waltham, MA, USA). PCR was performed using the following primers: 16S V3 (5′-TCG TCG GCA GCG TCA GAT GTG TAT AAG AGA CAG CCT ACG GGN GGC WGC AG-3′) and 16S V4 (5′-GTC TCG TGG GCT CGG AGA TGT GTA TAA GAG ACA GGA CTA CHV GGG TAT CTA ATC C-3′). The PCR cycle was as follows: initial denaturation for 2 min at 95 °C, followed by 30 cycles of denaturation for 20 s at 95 °C, annealing for 15 s at 72 °C, extension for 1 min at 72 °C, and a final extension step for 5 min.

Total DNA extracted from the kimchi samples was subjected to PCR using the 16S V4 primer. Sequencing was conducted using the Mi-Seq[TM] platform (Illumina, San Diego, CA, USA) by Macrogen (Macrogen Inc., Seoul, Korea). After sequencing errors and ambiguous and chimeric sequences were eliminated, the CD-HIT-OTU analysis program was used to determine the species-level operational taxonomic units (OTUs) to cluster sequences with a similarity of 97%. The representative OTU sequence was used to perform UCLUST (v.1.2.22) in the reference database (SIVA DB) and to generate taxonomic assignments based on homology. Microbial communities were analyzed using classifiers of the Ribosomal Database Project in QIME (v.1.8.0).

### 2.4. Analysis of pH and Titratable Acidity

The juice from the kimchi samples was extracted using a gauze after blending, and the pH value was measured directly using a pH meter (TitroLine 5000, SI Analytics GmbH, Mainz, Germany) at room temperature (24–26 °C). The titratable acidity was determined by titration with 0.1 N NaOH until the endpoint of pH 8.3 was reached. The titratable acidity was calculated based on the percentage of lactic acid produced [19].

### 2.5. Analysis of Organic Acids

Organic acid contents were measured using a modified procedure of a previous study [20]. Distilled water (50 mL) was added to 2 g of each sample, and the organic acids were extracted using a sonicator (PowerSonic 520, Hwashin Tech Co., Daegu, Korea) for 30 min. Subsequently, the solutions were filtered twice, first using a filter paper (Advantec No. 1, Toyo Roshi Kaisha, Ltd., Tokyo, Japan) and then using a syringe (Minisart RC, Hydrophilic, 0.2 μm, 15 mm, Sartorius Stedim Biotech GmbH, Goettingen, Germany). HPLC was conducted using an Agilent 1260 infinity/G4212B system (Agilent Technologies, Santa Clara, CA, USA) with a variable wavelength diode array detector set to 210 nm. The injection volume was 10 μL. Organic acids were analyzed using an Aminex HPX-87H column (300 × 7.8 mm, 9 μm, Bio-Rad, Hercules, CA, USA) maintained at 50 °C. Isocratic elution was performed using 0.008 N $H_2SO_4$, with deionized water as the mobile phase, for 30 min at a flow rate of 0.6 mL/min. Organic acids in the samples were identified by comparing their retention times with those of standard organic acids and quantified using a calibration curve derived from the peak areas of the standards.

### 2.6. Analysis of Free Sugars

The free sugar content of each sample was determined using a modified procedure from the previous study [21]. Ten grams of each homogenized kimchi sample was added to a 50 mL centrifuge tube, and adjustment to a constant volume was performed by adding 40 mL distilled water. The sample solution (50 mL) was heated in a water bath at 85 °C for 25 min and then cooled to room temperature. After centrifugation at 3000 rpm for 10 min, 1 mL of the supernatant was collected. The supernatant was filtered using a nylon membrane filter (0.45 μm, 25 mm, PTFE, Whatman GmbH, Dassel, Germany), and 6 μL of the supernatant was injected for the analysis of free sugars. The free sugar contents were measured by HPLC (1260 Infinity, Agilent Technologies, Santa Clara, CA, USA) using an instrument equipped with refractive index detectors. A carbohydrate column (Asahipak NH2P-50 4E, Shodex, Tokyo, Japan) was used at an oven temperature of 30 °C. The mobile phase was composed of 75% acetonitrile in water, dispensed at a flow rate of 1 mL/min. The free sugar content was estimated using standard curves of fructose, glucose, sucrose, and mannitol.

### 2.7. Analysis of Free Amino Acids

Free amino acids were measured using a modified procedure of a previous study [22]. For the analysis of free amino acids, 1 g of each homogenized kimchi sample was added to a 50 mL centrifuge tube, and adjustment to a constant volume was performed by adding 10 mL of distilled water. After centrifugation at 3000 rpm for 30 min, 1 mL of 5%

trichloroacetic acid was added to 1 mL of the supernatant and centrifuged at 6800 rpm for 20 min, following which the supernatant was collected. The supernatant was filtered using a syringe filter (RC, 0.2 μm, 25 mm, Sartorius) and analyzed using an automatic amino acid analyzer (L-8900, Hitachi, Tokyo, Japan). An ion-exchange column (4.6 × 60 nm, Hitachi HPLC Pack Column, #2622SC PF Column) was used for analysis, and the amino acids in the sample were detected using a UV detector (570 and 440 nm). For the mobile phase analysis, the Wako L-8900 buffer solution, dispensed at a flow rate of 0.35 mL/min, was used, and the samples were analyzed using a 20 μL sample.

### 2.8. Statistical Analysis

Statistical analyses were performed using GraphPad Prism 9.0 (GraphPad Software Inc., San Diego, CA, USA). For positive and negative correlation analyses between the bacterial community composition and the metabolites present in kimchi, a correlation analysis, using XLSTAT Premium Software Package version 19.4 (Addinsoft, New York, NY, USA), was performed based on the bacterial abundance in kimchi at the LAB species level and the targeted metabolite concentration. Correlation coefficient analysis was performed based on the relative abundances of the bacterial taxa (top six taxa with >1% of the mean abundance) at the species level and the kimchi metabolite contents. The statistical significance of the observed variation was assessed using the PERMANOVA function (* $p < 0.05$, ** $p < 0.01$, and *** $p < 0.001$). Principal component analysis (PCA), which is an unmanaged pattern recognition method, was used to identify the outliers in data sets and the storage periods. Correlation analysis between the PCA data and variables was performed using the XLSTAT.

## 3. Results and Discussion

### 3.1. Microbial Community Analysis

To compare the changes in the microbial community composition in the samples, the taxonomic structures at the species level were identified (Figure 1). In the initial stage of fermentation, the microbial communities in the five kimchi samples were similar at the species levels; *Aerosakkonema funiforme* and *Weisella confusa* were the predominant microorganisms in all samples. As fermentation progressed, the predominance of *Latilactobacillus sakei* and *Leuconostoc gelidum* was observed. Interestingly, in the first week of fermentation, *L. gelidum* was the predominant microorganism, and its ratio was high in SK-A with low salinity (43.69%). Conversely, *A. funiforme* was the primary microorganism in kimchi with high salinity (SK-E). This result was similar to that obtained in a previous study that confirmed the changes in microbial community composition at high salinity; the authors reported an increase in the abundance of *A. funiforme* and a decrease in that of *L. gelidum* [5]. After 5 weeks, the abundance of *L. gelidum* reduced, whereas that of *L. sakei* showed an increasing trend. In particular, *L. sakei* was the predominant microorganism in kimchi samples with high salinity, in contrast to that in other samples. These results indicated that *Leuconostoc* and *Latilactobacillus* play an important role in kimchi fermentation. During the initial stage of fermentation, when the kimchi has low levels of aeration and weakly acidic conditions, *Leuconostoc* species are the predominant microbial species present; as fermentation progresses, the acidity increases, and an anaerobic environment is established [23,24]. Because *Latilactobacillus* grow and adapt well under highly acidic and anaerobic conditions, the rapid increase in acidity and the establishment of anaerobic conditions toward the end of fermentation are advantageous for its growth. This is consistent with the results of previous studies [25].

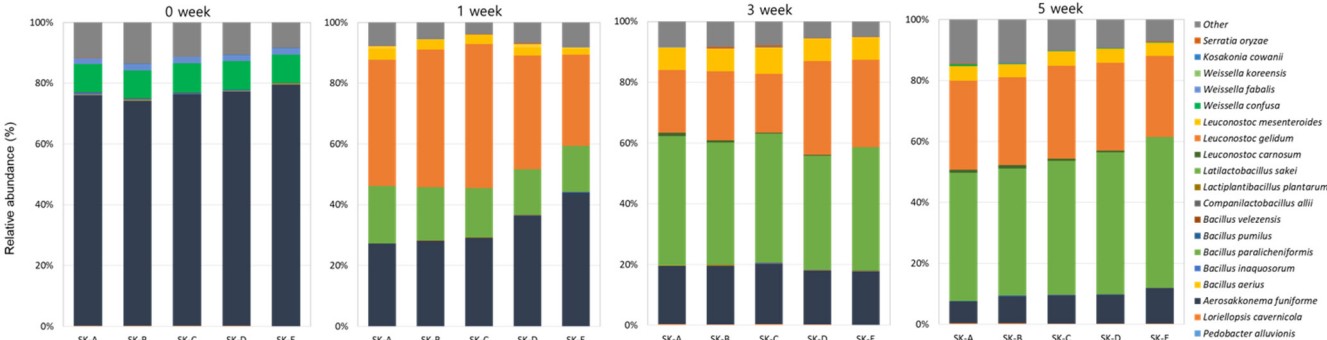

**Figure 1.** Bacterial community composition at the species level in kimchi fermented at different salinities, as found using the SILVA rRNA database. 'Other' is composed of genera among which each shows a percentage of reads <0.5% of the total reads in all kimchi samples in species-level analyses.

### 3.2. Changes in the pH and Titratable Acidity of Kimchi

The pH changes in kimchi at different salinities were analyzed for 5 weeks (Figure 2a). pH and titratable acidity are important factors in kimchi fermentation [26]. On the first day of fermentation, the pH was approximately 5.4. There were no differences in the pH values between the different groups of kimchi. A gradual decrease in pH was observed as fermentation progressed. After 1 week, the pH value started decreasing; in particular, the pH value of SK-A, which had low salinity, declined sharply. After 2 weeks, the pH of the other kimchi samples also declined sharply, and there were no significant differences among the pH values of the different kimchi samples at the end of the fermentation period. Hong et al. [16] reported that the pH of kimchi with low salinity was lower than that of kimchi with high salinity at the initial stage, and the pH value declined rapidly during the initial period of fermentation. In the early stages of fermentation, the pH declined sharply, but the decline was gradual after the midpoint of the fermentation period; meanwhile, the acidity was reported to increase continuously, and the pH was further reduced by the organic acids produced by LAB [20]. The titratable acidities of kimchi during fermentation are shown in Figure 2b. In general, titratable acidity is inversely proportional to the pH value. Our results indicated that the titratable acidity of kimchi was significantly affected by its salinity. Overall, the titratable acidity of all kimchi samples increased rapidly during the 5 weeks of fermentation. The titratable acidity increased slightly and became relatively stable, with values ranging from 0.5 to 1.0, after 2 weeks. Salinity was shown to lower the initial titratable acidity of kimchi, which may have further delayed kimchi fermentation at the initial stage. The rapid decrease in pH observed in this study was slightly different from what is generally observed in kimchi fermentation. The pH and titratable acidity of kimchi with different salinities were similar to those generally observed in kimchi fermentation during the 5-week fermentation period [27]. Till the second week of fermentation, the titratable acidity of the high-salinity kimchi group was significantly lower than that of the low-salinity kimchi group. Similarly, Hong et al. [16] reported that at the initial stage, the pH of kimchi with low salinity was lower than that of kimchi with high salinity, and the pH value decreased rapidly during the initial period of fermentation.

### 3.3. Changes in the Metabolite Profile of Kimchi

It is well known that the taste and flavor of kimchi are primarily related to its metabolite profile, and metabolite production is affected by the microbial community composition during kimchi fermentation. To investigate the effect of salinity on kimchi fermentation, the kimchi metabolites were analyzed using HPLC with a metabolomics approach.

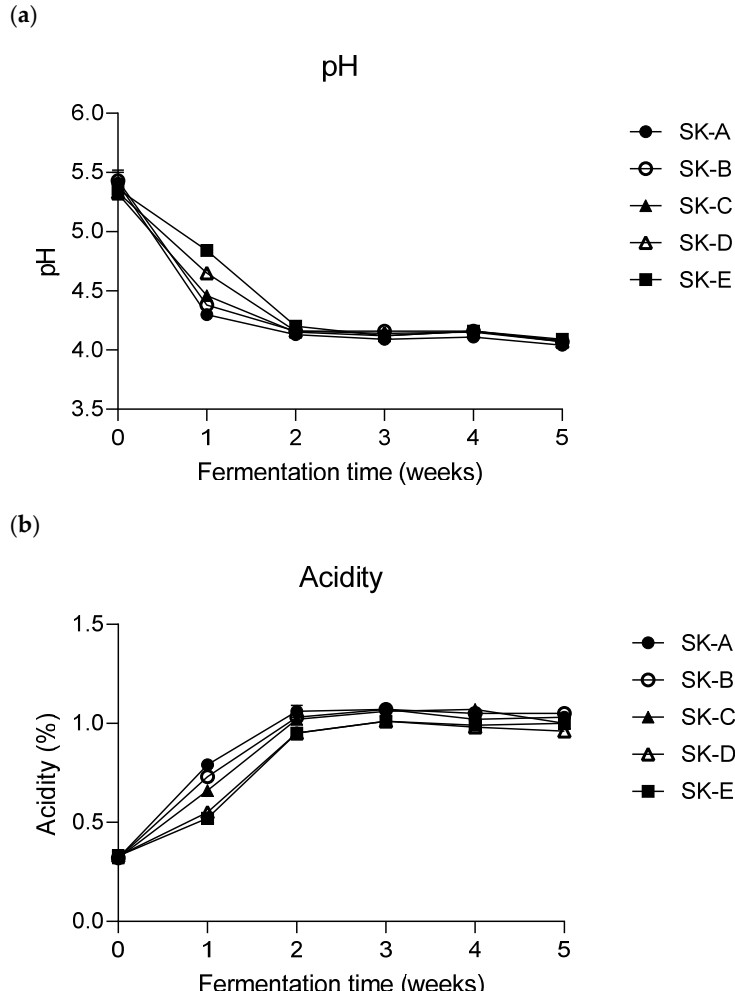

**Figure 2.** Changes in the pH (**a**) and acidity (**b**) of kimchi samples with different salinities during fermentation.

The changes in the organic acid content of kimchi samples fermented for 5 weeks at different salinities at 6 °C are presented in Figure 3. Six organic acids (acetic acid, citric acid, malic acid, succinic acid, lactic acid, and fumaric acid) were detected during kimchi fermentation. The lactic acid levels increased sharply in the first 2 weeks and increased gradually after 2 weeks. In particular, the level of lactic acid was significantly higher in low-salinity kimchi samples during the first and second weeks of fermentation. The acetic acid levels increased steadily for 5 weeks, and kimchi samples with low salinity eventually showed a high acetic acid content. Lactic acid is an organic acid whose content increases during fermentation [11]. The change in titratable acidity is similar to the change in the lactic acid content, and it seems that the titratable acidity of kimchi is primarily affected by lactic acid; therefore, lactic acid is an important indicator of the quality of kimchi [28]. In this study, the change in titratable acidity and the change in lactic acid content showed similar tendencies, and the titratable acidity of kimchi was primarily observed to be affected by lactic acid. Malic acid was produced during the initial stage of fermentation, but its concentration decreased to zero by the end of the fermentation period. According to Shim et al. [29], malic acid is commonly converted to lactic acid and acetic acid by LAB during fermentation. The lactic acid content increased during fermentation, as it was the primary organic acid, whereas the malic acid content decreased steadily. Seo et al. [9] also showed that the acetic acid and lactic acid contents increased during fermentation, whereas the succinic acid and malic acid contents decreased.

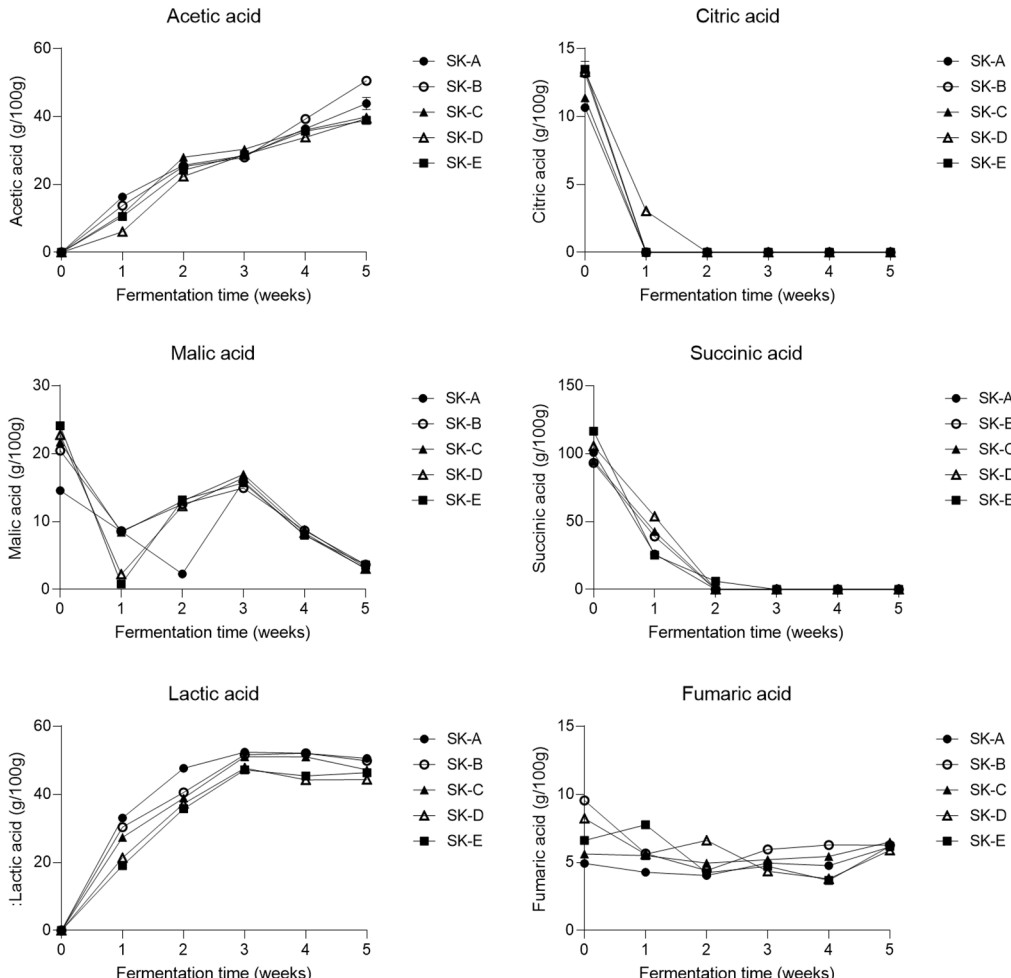

**Figure 3.** Changes in the organic acid contents of kimchi samples with different salinities during fermentation.

The changes in the free sugar content of kimchi samples fermented at different salinities at 6 °C for 5 weeks are presented in Figure 4. In general, the free sugars primarily detected were fructose, glucose, maltose, sucrose, and mannitol, and the free sugar concentration in low-salt kimchi decreased in the early stage of fermentation. The free sugar content in all kimchi groups decreased during the fermentation process; a similar finding was reported in a previous study [3,20], in which the free sugar contents in all treatment groups decreased rapidly during the initial stage of fermentation. The decrease in the levels of free sugar, especially glucose, could be presumably attributed to sugar consumption by the rapidly proliferating LAB present in kimchi rather than to the release of free sugars in the cabbage. In this study, the fructose and glucose levels decreased rapidly until 2 weeks, but the fructose level declined more rapidly, and fructose was completely consumed during the 2 weeks.

Mannitol, a sugar alcohol, was not detected at the beginning of the fermentation period, but its production increased to a significant level in all kimchi samples after 1 week of fermentation, and the level increased significantly after 2 weeks of fermentation. In particular, the level of mannitol was high in low-salinity samples in the first and second weeks of fermentation. During this period, the *Leuconostoc* species ratio in the low-salinity kimchi group (SK-A) was relatively higher than that in the high-salinity kimchi group (SK-E). *L. mesenteroides* along with other heterofermentative LAB is known to produce high levels of mannitol from a mixture of glucose and fructose [21,24,30]. Therefore, it is considered that kimchi with a high ratio of *Leuconostoc* produces mannitol at high levels.

The changes in the free amino acid composition of kimchi samples fermented at different salinities at 6 °C for 5 weeks are presented in Figure 5. The concentrations of most amino acids produced in kimchi are maintained or increased during fermentation. Jung et al. [23] reported that the concentration of amino acids increased rapidly at the beginning of fermentation and gradually decreased after the fermentation was halfway through. Similarly, in this study, the glutamic acid content decreased rapidly at the beginning of fermentation and then increased, as also reported by Park et al. [27]. In contrast, the concentrations of glycine, methionine, cystathionine, leucine, tyrosine, γ-amino-n-butyric acid (GABA), and lysine increased during fermentation, and there was no significant difference in amino acid concentrations based on salinity.

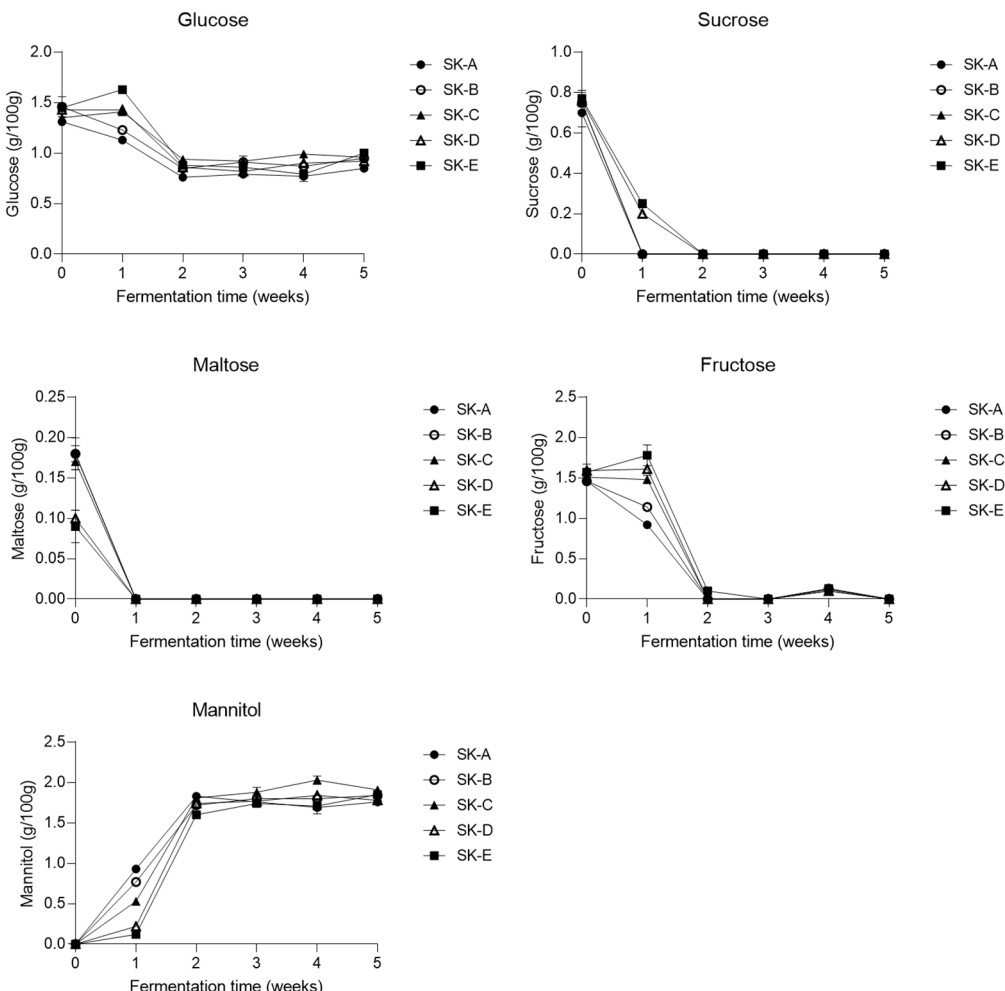

**Figure 4.** Changes in the free sugar contents of kimchi samples with different salinities during fermentation.

### 3.4. Multivariate Statistical Analysis

PCA is a popular technique of multivariate statistical analysis used to simplify data from multidimensional datasets that can be interpreted through graphical visualization [31]. The shift of the sample (dots) from left to right in the PCA score plot indicated continuous metabolic changes during fermentation (Figure 6a). After 2 weeks of fermentation, till the 5th week of fermentation, the shift of the sample proceeded slowly, indicating that the change in the metabolite profile was slow during this period, which is consistent with the findings from previous studies on varying salinities in kimchi fermentation [4,9]. To determine the differences in the metabolite profiles of kimchi samples with different salinities during 5 weeks of fermentation, a PCA model was constructed using variables obtained from free sugar, organic acid, and free amino acid data at 0, 1, 3, and 5 weeks.

At the beginning of the fermentation period, kimchi with various salinities could not be completely distinguished on the PCA score plot, but during the first week of fermentation, the kimchi samples could be separated according to salinity.

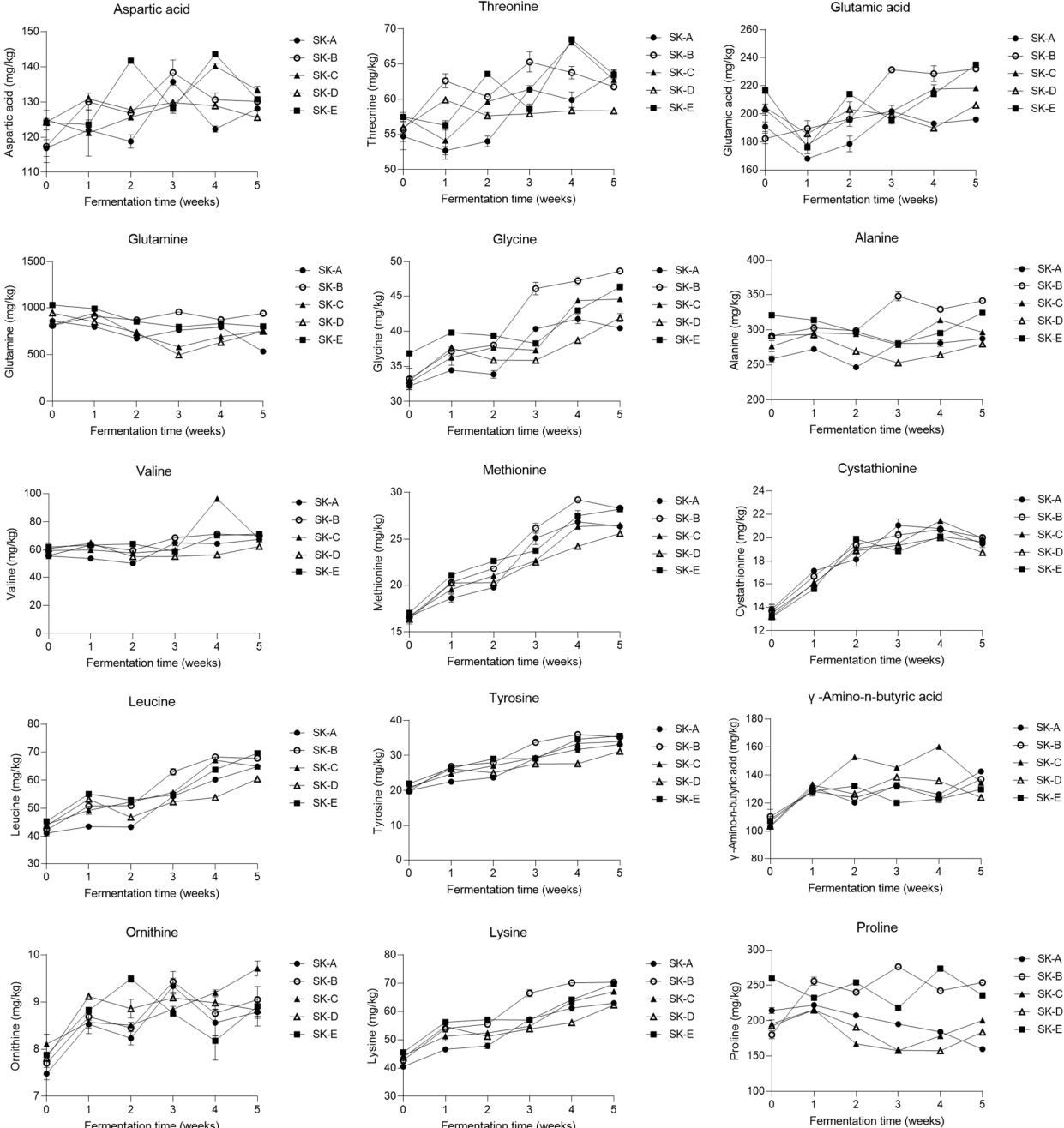

**Figure 5.** Changes in the free amino acid contents of kimchi samples with different salinities during fermentation.

(a)

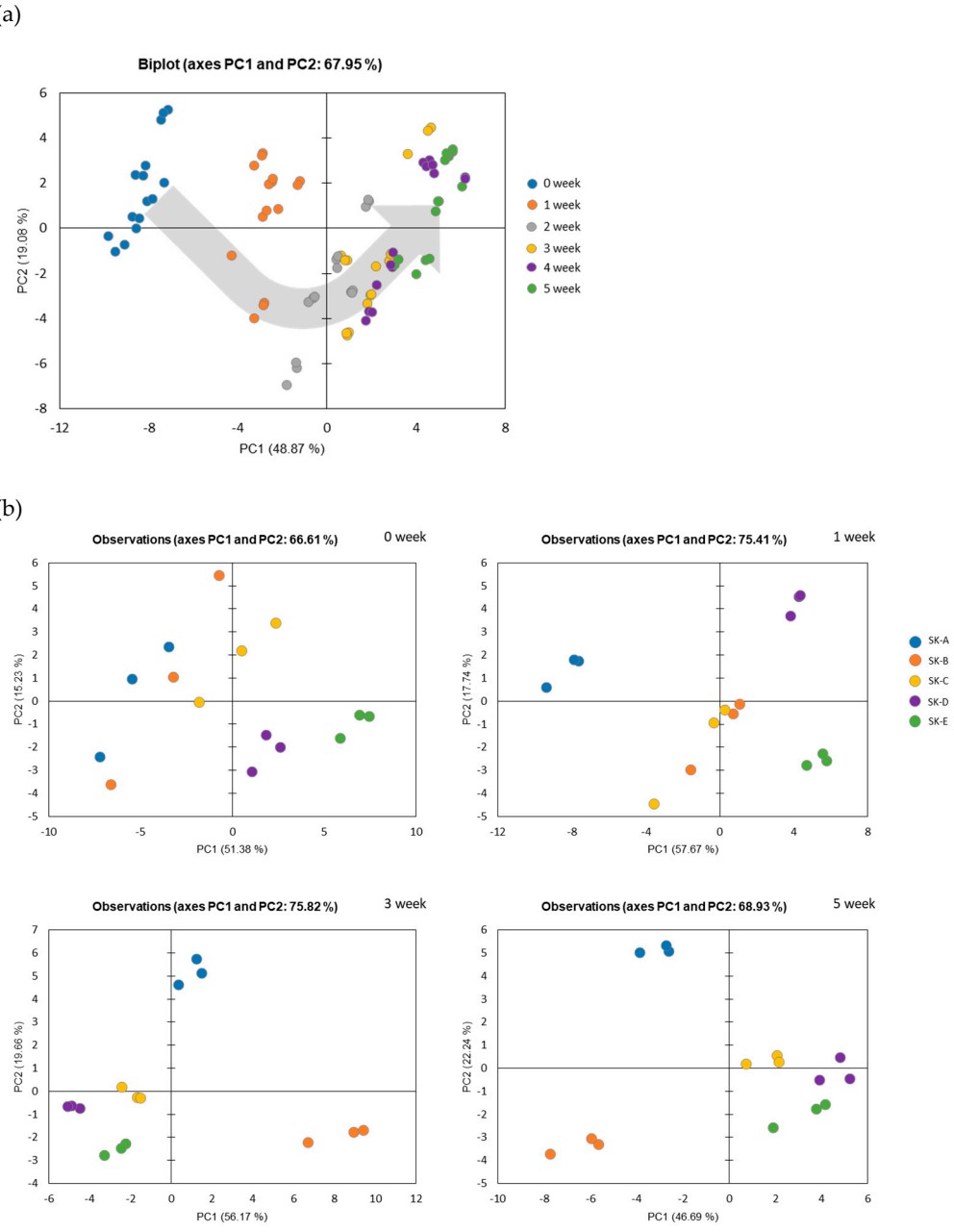

**Figure 6.** PCA score plots derived from metabolite data of kimchi samples fermented at different salinities. (**a**) PCA score plots derived from metabolite data for each day of fermentation. (**b**) PCA score plots derived from metabolite data for kimchi with different salinities at 0, 1, 3, and 5 weeks of fermentation. PCA, principal component analysis.

As shown in Figure 6b, the metabolite profiles at different salinities could be completely distinguished from each other from the third week of fermentation. At the third week of fermentation, the samples were completely distinguishable on the PCA score plot, and in particular, SK-A and SK-B could be clearly distinguished till the 5th week of fermentation. Positive and negative correlations between species-level transitions and changes in the metabolite profile during kimchi fermentation were analyzed and presented as a heatmap (Figure 7). Correlation coefficient analysis was performed using the relative abundance of abundant bacterial taxa (top six taxa with >1% of the mean abundance) at the species level and the abundance profiles of kimchi metabolites. A similar heatmap pattern was observed for samples with a limited difference in salinity, whereas the two kimchi samples with a substantial difference in salinity, SK-A and SK-E, showed markedly differ-

ent patterns, particularly with respect to amino acid metabolites. The negative correlation between amino acid concentrations and LAB species was higher in high-salinity kimchi. The changes owing to salinity may be related to the amino acid predominantly consumed by LAB. More specifically, it was shown that the abundances of *L. sakei*, *L. carnosum*, and *L. mesenteroides* were significantly positively correlated with the acetic acid and lactic acid contents and negatively correlated with the glucose and fructose contents. The levels of methionine, leucine, cystathionine, and GABA showed a significant positive correlation with the abundances of *L. sakei* and *L. mesenteroides*. Kimchi fermented with fermented fish sauce showed higher levels of acetate, lactate, and GABA than kimchi fermented without fermented fish sauce [32]. The mannitol content showed a significant positive correlation with the abundances of *L. sakei* and *L. mesenteroides* ($p < 0.001$). Kim et al. [16] reported that *L. sakei* can produce mannitol at low concentrations because of its homofermentative and heterofermentative characteristics. Another study showed the production mannitol at low levels by some homofermentative LAB, such as *Lactobacillus leichmannii* and *Lactococcus lactis* [22]. Therefore, PCA and heatmap are suitable statistical methods for evaluating the correlation between salinity and the different characteristics of kimchi. Kimchi with different quality characteristics can be produced by varying the salinity ratio, which is in agreement with the results of previous studies [7,8,14].

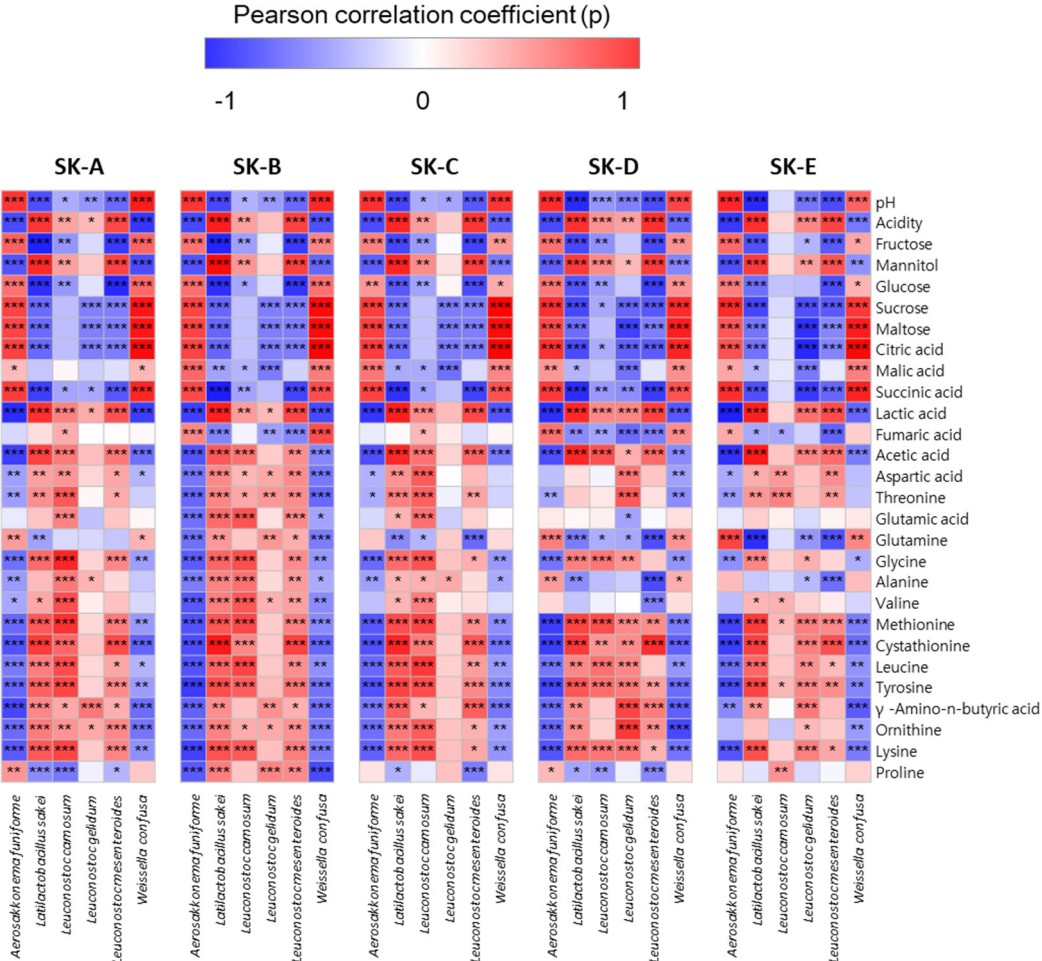

**Figure 7.** Correlation analysis between successive colonization by lactic acid bacteria and metabolite profiles in kimchi samples fermented at different salinities. The blue and red colors correspond to negative and positive correlation, respectively. The color intensity is proportional to the correlation coefficient. The color bar indicates the corresponding color and correlation coefficient. * $p < 0.05$, ** $p < 0.01$, *** $p < 0.001$.

## 4. Conclusions

In this study, the changes in the characteristics of kimchi at different salinities were observed, and the correlation between the types of LAB and the metabolites produced was analyzed. Interestingly, the abundances of microbial communities and concentrations of metabolites in kimchi differed according to the salinity. The results of metabolite analysis, including multivariate statistical data of organic acids, free sugars, and free amino acids, and the microbial community analysis data, providing insights into the unique characteristics of kimchi fermented at different salinities. The results obtained in this study can be applied to fermented foods processed at different salt concentrations. Appropriate salt concentrations can alter the microbial community composition and metabolite profile in a specific food product, which can positively affect its quality.

**Author Contributions:** Conceptualization, H.-Y.S. and L.-S.K.; methodology, H.-Y.S., S.M., S.-H.P., and Y.B.C.; validation; Y.-R.Y.; formal analysis, Y.-J.C., S.J.P., H.J.L., S.H., and H.L.; investigation, M.-A.L. and Y.-J.C.; data curation, M.-A.L.; writing-original draft preparation, M.-A.L.; writing-review and editing, H.-Y.S. and L.-S.K.; visualization, Y.-J.C.; supervision, H.-Y.S.; funding acquisition, H.-Y.S. All authors have read and agreed to the published version of the manuscript.

**Funding:** This research was supported by a grant from the World Institute of Kimchi (KE2102-2-2), funded by the Ministry of Science and ICT, Republic of Korea.

**Institutional Review Board Statement:** Not applicable.

**Informed Consent Statement:** Not applicable.

**Data Availability Statement:** The data presented in this study are available in the article.

**Conflicts of Interest:** The authors declare no conflict of interest.

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
