# Peer review of "Influence of Salinity on the Microbial Community Composition and Metabolite Profile in Kimchi"

_fermentation, doi:10.3390/fermentation7040308_

Round 1
Reviewer 1 Report
The authors should clearly present the novelty of their work since several similar studies are available in the literature.
Some of them are presented by the authors like
Kim, D. W., Kim, B. M., Lee, H. J., Jang, G. J., Song, S. H., Lee, J. I., ... & Kim, H. J. (2017). Effects of different salt treatments on the fermentation metabolites and bacterial profiles of kimchi. Journal of food science, 82(5), 1124-1131. https://doi.org/10.1111/1750-3841.13713
Lee, K. W., Shim, J. M., Kim, D. W., Yao, Z., Kim, J. A., Kim, H. J., & Kim, J. H. (2018). Effects of different types of salts on the growth of lactic acid bacteria and yeasts during kimchi fermentation. Food science and biotechnology, 27(2), 489-498. https://doi.org/10.1007/s10068-017-0251-7
Hong, G.H.; Lee, S.Y.; Park, E.S.; Park, K.Y. Changes in Microbial Community by Salt Content in Kimchi During Fermentation. J. Korean Soc. Food Sci. Nutr. 2021, 50 (6), 648–653. https://doi.org/10.3746/jkfn.2021.50.6.648
Especially the last one is very similar with the present study even at Korean language.!!!
However, there are a number of similar studies that are not reported by the authors. Some examples:
Sui, M., Qu, P., Zhu, Y., Zhang, F., & Li, C. (2019, October). Optimization of the Fermentation Process of Kimchi. In IOP Conference Series: Materials Science and Engineering (Vol. 612, No. 2, p. 022040). IOP Publishing. https://iopscience.iop.org/article/10.1088/1757-899X/612/2/022040/meta
Lee, M., Song, J. H., Jung, M. Y., Lee, S. H., & Chang, J. Y. (2017). Large-scale targeted metagenomics analysis of bacterial ecological changes in 88 kimchi samples during fermentation. Food microbiology, 66, 173-183. https://doi.org/10.1016/j.fm.2017.05.002
There is also a similar study since 1984!! With the effect of salt concentration in microbial community of Kimchi Mheen, T. I., & Kwon, T. W. (1984). Effect of temperature and salt concentration on kimchi fermentation. Korean Journal of Food Science and Technology, 16(4), 443-450. https://www.koreascience.or.kr/article/JAKO198403041899994.page
Finally there is a study available with the effect of salt content of organic acids of Kimchi Seo, S. H., Park, S. E., Kim, E. J., Lee, K. I., Na, C. S., & Son, H. S. (2018). A GC-MS based metabolomics approach to determine the effect of salinity on Kimchi. Food Research International, 105, 492-498. https://doi.org/10.1016/j.foodres.2017.11.069
Author Response
Reviewer #1: The authors should clearly present the novelty of their work since several similar studies are available in the literature.
Some of them are presented by the authors like
Kim, D. W., Kim, B. M., Lee, H. J., Jang, G. J., Song, S. H., Lee, J. I., ... & Kim (2017). Effects of different salt treatments on the fermentation metabolites and bacterial profiles of kimchi. Journal of food science, 82(5), 1124-1131. https://doi.org/10.1111/1750-3841.13713
Lee, Shim, Kim, Yao, Kim, Kim, & Kim (2018). Effects of different types of salts on the growth of lactic acid bacteria and yeasts during kimchi fermentation. Food science and biotechnology, 27(2), 489-498. https://doi.org/10.1007/s10068-017-0251-7
→ The previous studies you have indicated above analyzed metabolites and bacterial profiles of kimchi according to the types of salt (purified salt, mineral rich sea salt, solar salt, and bamboo salt). Our study analyzed the effect of the salt concentration of kimchi.
Hong, G.H.; Lee, S.Y.; Park, E.S.; Park, K.Y. Changes in Microbial Community by Salt Content in Kimchi During Fermentation. J. Korean Soc. Food Sci. Nutr. 2021, 50 (6), 648–653. https://doi.org/10.3746/jkfn.2021.50.6.648.
Especially the last one is very similar with the present study even at Korean language.!!!
→ Our study differs from the above previous study as follows: Our preliminary study for monitoring the salinity of kimchi distributed on the market revealed that the salt concentration of kimchi was 1.4~2.5%. Based on this, this study aimed to analyze the changes in metabolites and microbial profiles by adjusting the salt concentration of kimchi up to 1.4, 1.7, 2.0, 2.2, and 2.5%, and analyzed the correlation accordingly. However, in the above previous study, only microbial community analysis was performed, and the correlation with metabolites was not analyzed.
However, there are a number of similar studies that are not reported by the authors. Some examples:
Sui, M., Qu, P., Zhu, Y., Zhang, F., & Li, C. (2019, October). Optimization of the Fermentation Process of Kimchi. In IOP Conference Series: Materials Science and Engineering (Vol. 612, No. 2, p. 022040). IOP Publishing. https://iopscience.iop.org/article/10.1088/1757-899X/612/2/022040/meta
→ The previous study (Sui et al., 2019) only used similar salt concentration conditions to our study, but its purpose is different from that of our study.
Lee, Song, Jung, Lee, & Chang (2017). Large-scale targeted metagenomics analysis of bacterial ecological changes in 88 kimchi samples during fermentation. Food microbiology, 66, 173-183. https://doi.org/10.1016/j.fm.2017.05.002
→ The above previous study (Lee et al. 2017) collected 88 kimchi samples and showed the results of different microbial communities depending on fermentation conditions, such as salt concentration, major ingredients, fermentation period, and sample collection time. Therefore, it is different from our study.
There is also a similar study since 1984!! With the effect of salt concentration in microbial community of Kimchi Mheen, & Kwon (1984). Effect of temperature and salt concentration on kimchi fermentation. Korean Journal of Food Science and Technology, 16(4), 443-450. https://www.koreascience.or.kr/article/JAKO198403041899994.page
→ Our preliminary study for monitoring the salinity of kimchi distributed on the market revealed that the salt concentration of kimchi was 1.4~2.5%. The salt concentration (2.25, 3.5, 5.0, 7.0%) suggested in the previous study (Mheen & Kwon, 1984) are for research purposes, and no kimchi is actually prepared in this manner. There is no kimchi that has such a high concentration of salt (3.5~7.0% salinity of kimchi). In addition, in our study, a correlation analysis between metabolites and microbial profiles according to salt concentration was conducted to differentiate it from the previous study.
Finally there is a study available with the effect of salt content of organic acids of Kimchi Seo, Park, Kim, Lee, Na, & Son (2018). A GC-MS based metabolomics approach to determine the effect of salinity on Kimchi. Food Research International, 105, 492-498. https://doi.org/10.1016/j.foodres.2017.11.069
→ Our preliminary study for monitoring the salinity of kimchi distributed on the market revealed that the salt concentration of kimchi was 1.4~2.5%. The salt concentration (0, 1.25, 2.5, 5%) suggested in the previous study (Park et al., 2018) is for research purposes, and no kimchi is actually prepared in this manner. There is no kimchi that has such a high concentration of salt (5% salinity of kimchi)
Reviewer 2 Report
Dear Authors,
The reviewed Ms ID: fermentation-1462162 (Influence of Salinity on the Microbial Community Composition and Metabolite Profile in Kimchi) is really absorbing and contributes to updates in literature by introducing more information on the effects of salinity on kimchi fermentation.
A great advantage is the clear aim of the study, the use of advanced research methods and the crystal clear results of the experiment, both in the text and in the figures presented. It is a great pleasure to read the text of this manuscript.
However, there are minor flaws of the manuscript need to be fixed. Specific comments on the manuscript are as follows:
- the lack of numbering of lines in the manuscript, makes it difficult to indicate where to make corrections
- in the first subparagraph of the section Introduction , please complete more references (after references No. 5-6). “Several studies have suggested that fermentation …”. It is the same in the second subparagraph (after references No. 11-12). ”However, several studies have suggested that high salt intake from dietary sources is strongly associated with hypertension and cardiovascular…”
- Subsection 2.4-2.7: please complete references,
- The presentation of the obtained results (especially Figure 1; Figure 7) is incredibly similar to the information given in previous article:
Hye Seon Song, Se Hee Lee, Seung Woo Ahn, Joon Yong Kim, Jin-Kyu Rhee, Seong Woon Roh, Effects of the main ingredients of the fermented food, kimchi, on bacterial composition and metabolite profile, Food Research International,Volume 149, 2021,110668, ISSN 0963-9969,https://doi.org/10.1016/j.foodres.2021.110668.
From my standpoint, this manuscript is appropriate for publication in Journal – Fermentation after minor revision, given the above aspects.
Author Response
The reviewed Ms ID: fermentation-1462162 (Influence of Salinity on the Microbial Community Composition and Metabolite Profile in Kimchi) is really absorbing and contributes to updates in literature by introducing more information on the effects of salinity on kimchi fermentation. A great advantage is the clear aim of the study, the use of advanced research methods and the crystal clear results of the experiment, both in the text and in the figures presented. It is a great pleasure to read the text of this manuscript. However, there are minor flaws of the manuscript need to be fixed. Specific comments on the manuscript are as follows:
the lack of numbering of lines in the manuscript, makes it difficult to indicate where to make corrections
in the first subparagraph of the section Introduction , please complete more references (after references No. 5-6). “Several studies have suggested that fermentation …”. It is the same in the second subparagraph (after references No. 11-12). ”However, several studies have suggested that high salt intake from dietary sources is strongly associated with hypertension and cardiovascular…”
Subsection 2.4-2.7: please complete references,
→ The manuscript has been revised.
The presentation of the obtained results (especially Figure 1; Figure 7) is incredibly similar to the information given in previous article:
Hye Seon Song, Se Hee Lee, Seung Woo Ahn, Joon Yong Kim, Jin-Kyu Rhee, Seong Woon Roh, Effects of the main ingredients of the fermented food, kimchi, on bacterial composition and metabolite profile, Food Research International, Volume 149, 2021,110668, ISSN 0963-9969,https://doi.org/10.1016/j.foodres.2021.110668.
→ The method for data presentation is similar, but the purpose is different from that of the previous study. The study by Song et al. (2021) showed the results of microbial profiles and metabolites analysis according to the type of main ingredients, and this paper showed that microbial profiles and metabolites vary depending on the salt concentration of kimchi.
Reviewer 3 Report
To determine the effect of Salinity on Kimchi fermentation, microbial Community and metabolites during the whole fermentation period were fully examined. The findings are beneficial for good manufacture practice of Kimchi production, and can also applied to other fermented foods. The manuscript is neatly writing, and the conclusions are well supported by the results. It can be accepted after the following comment.
Comment:
The caption of figure 6 should indicate what figure 6a and figure 6b means.
Author Response
To determine the effect of Salinity on Kimchi fermentation, microbial Community and metabolites during the whole fermentation period were fully examined. The findings are beneficial for good manufacture practice of Kimchi production, and can also applied to other fermented foods. The manuscript is neatly writing, and the conclusions are well supported by the results. It can be accepted after the following comment.
Comment:
The caption of figure 6 should indicate what figure 6a and figure 6b means.
→ The manuscript has been revised.